# Combining Three Cohorts of World Trade Center Rescue/Recovery Workers for Assessing Cancer Incidence and Mortality

**DOI:** 10.3390/ijerph18041386

**Published:** 2021-02-03

**Authors:** Robert M. Brackbill, Amy R. Kahn, Jiehui Li, Rachel Zeig-Owens, David G. Goldfarb, Molly Skerker, Mark R. Farfel, James E. Cone, Janette Yung, Deborah J. Walker, Adrienne Solomon, Baozhen Qiao, Maria J. Schymura, Christopher R. Dasaro, Dana Kristjansson, Mayris P. Webber, Roberto G. Lucchini, Andrew C. Todd, David J. Prezant, Paolo Boffetta, Charles B. Hall

**Affiliations:** 1World Trade Center Health Registry, New York City Department of Health and Mental Hygiene, Long Island City, NY 11101, USA; jli3@health.nyc.gov (J.L.); mfarfel@health.nyc.gov (M.R.F.); jcone@health.nyc.gov (J.E.C.); jyung@health.nyc.gov (J.Y.); dwalker@health.nyc.gov (D.J.W.); asolomon1@health.nyc.gov (A.S.); 2Bureau of Cancer Epidemiology, New York State Department of Health, Albany, NY 12237, USA; amy.kahn@health.ny.gov (A.R.K.); baozhen.qiao@health.ny.gov (B.Q.); maria.schymura@health.ny.gov (M.J.S.); 3Fire Department of the City of New York (FDNY), Brooklyn, NY 11201, USA; Rachel.Zeig-Owens@fdny.nyc.gov (R.Z.-O.); David.Goldfarb@fdny.nyc.gov (D.G.G.); Molly.Skerker@fdny.nyc.gov (M.S.); Mayris.Webber@fdny.nyc.gov (M.P.W.); David.Prezant@fdny.nyc.gov (D.J.P.); 4Department of Medicine, Montefiore Medical Center, New York, NY 10467, USA; 5Department of Epidemiology and Population Health, Albert Einstein College of Medicine, Bronx, NY 10461, USA; Charles.Hall@einsteinmed.org; 6Department of Environmental Medicine and Public Health, Icahn School of Medicine at Mount Sinai, New York, NY 10029, USA; christopher.dasaro@mssm.edu (C.R.D.); roberto.lucchini@mssm.edu (R.G.L.); andrew.todd@mssm.edu (A.C.T.); 7Department of Genetics and Bioinformatics, Norwegian Institute of Public Health, 0456 Oslo, Norway; danakristjansson@gmail.com; 8Stony Brook Cancer Center, Stony Brook University, Stony Brook, NY 11794, USA; Paolo.Boffetta@stonybrookmedicine.edu; 9Department of Medical and Surgical Sciences, University of Bologna, 40138 Bologna, Italy

**Keywords:** World Trade Center, exposure, cancer, rescue and recovery workers, pooling cohorts

## Abstract

Three cohorts including the Fire Department of the City of New York (FDNY), the World Trade Center Health Registry (WTCHR), and the General Responder Cohort (GRC), each funded by the World Trade Center Health Program have reported associations between WTC-exposures and cancer. Results have generally been consistent with effect estimates for excess incidence for all cancers ranging from 6 to 14% above background rates. Pooling would increase sample size and de-duplicate cases between the cohorts. However, pooling required time consuming steps: obtaining Institutional Review Board (IRB) approvals and legal agreements from entities involved; establishing an honest broker for managing the data; de-duplicating the pooled cohort files; applying to State Cancer Registries (SCRs) for matched cancer cases; and finalizing analysis data files. Obtaining SCR data use agreements ranged from 6.5 to 114.5 weeks with six states requiring >20 weeks. Records from FDNY (*n* = 16,221), WTCHR (*n* = 29,372), and GRC (*n* = 33,427) were combined de-duplicated resulting in 69,102 unique individuals. Overall, 7894 cancer tumors were matched to the pooled cohort, increasing the number cancers by as much as 58% compared to previous analyses. Pooling resulted in a coherent resource for future research for studies on rare cancers and mortality, with more representative of occupations and WTC- exposure.

## 1. Introduction

The attacks on the World Trade Center (WTC) in New York on 11 September 2001 resulted in the deaths of almost 3000 civilians and responders who were occupants of the WTC buildings or in their immediate vicinity when the Twin Towers collapsed. Further, from the morning of 9/11 and continuing for nine or more months, up to 90,000 rescue/recovery workers (i.e., first responders, rescue, recovery and clean-up workers) [1] were exposed to a wide range of potentially biologically active and hazardous substances. These included pulverized cement, glass, asbestos, lead, polycyclic aromatic hydrocarbons), polychlorinated biphenyls, organochlorine pesticides, and polychlorinated furans and dioxins [2,3]. These exposures may have increased the responders’ risk of developing various conditions and diseases, including cancer.

Since the WTC attacks, several programs have devoted considerable effort to identify health effects resulting from WTC-exposure among rescue/recovery workers. These programs include the Fire Department of the City of New York (FDNY), which houses the FDNY WTC Health Program; the New York City Department of Health and Mental Hygiene, which maintains the World Trade Center Health Registry (WTCHR); and, the CDC/NIOSH WTC Health Program General Responder Clinical Centers of Excellence (Rutgers University, New York University, Icahn School of Medicine at Mount Sinai [ISMMS], Queens College/Northwell Health, and the State University of New York) and Data Center (GRDC, at ISMMS) for the General Responder Cohort (GRC).

Each of these programs has published one or more studies on associations between WTC-exposure and cancer. In general, the results have been consistent, with effect estimates for all cancers from the most recent studies ranging from 6 to 14 % in excess of background rates [4,5,6,7]. The effect for all cancers was not statistically significant in all studies but was for selected cancer sites [4,5,6,7].

An overview of the challenges of conducting surveillance based on separate cohorts provided a roadmap for the creation of a joint study using pooled data across these cohorts [8]. However, there are important differences between the cohorts. The WTCHR has a fully closed cohort (i.e., after the end of enrollment, the cohort is fixed and follow-up time can be estimated), FDNY has a mostly closed cohort and the GRC is an open cohort that continues to enroll responders into a member-centered medical monitoring and treatment program. The cohorts also vary in the occupations represented: the FDNY cohort includes only firefighters and Emergency Medical Service Providers (EMS) employed by NYC, but the WTCHR and GRC both include any persons who performed paid or volunteer rescue or recovery work. Further, participants can be in more than one of these WTC-exposed cohorts.

Notwithstanding the differences in design and composition of these cohorts, there are substantial benefits to pooling the data from the cohorts for analysis after matching and de-duplication. Many of the advantages and disadvantages of this approach have been outlined by Boffetta, et al., 2016 [8]. The most important advantage is the increase in sample size and resultant improvement in statistical power, especially for less common types of cancers, such as hematologic cancers. Additional advantages include: (a) increasing the heterogeneity of occupations thereby providing a greater degree of generalizability of the findings; (b) improving consistency and efficiency by using one application for matching to cancer registries and obtaining linkage results; (c) eliminating bias in meta-analysis through de-duplication of cases across cohorts; (d) using a common method for defining level of 9/11 exposure; (e) applying the same reference population for a specific set of years for computing expected rates; and, (f) adjusting for differences in results for cohort design effects by analyzing pooled data.

Pooling data across the three cohorts also presents analytical challenges. For example, there are different levels of detail on self-reported exposure to 9/11 provided by the cohorts. As such, the WTCHR has detailed exposure assessment providing both time of arrival and duration of work at the WTC site [on the pile (area where debris from WTC towers’ collapse was most concentrated) or not on the pile], kinds of tasks performed at the site, and exposure to the dust cloud. The FDNY has information on dates and times members were at the WTC site and on the tasks performed there. The GRC has some of the information noted above in addition to the degree of exposure to the dust cloud on 9/11 at time of the building collapses.

The current project, “Incidence, Latency, and Survival after World Trade Center Exposure,” was funded by the National Institute for Occupational Safety and Health (NIOSH) in 2016 to establish the “Combined WTC Rescue/recovery Cohort” (the combined Cohort), a study population that includes rescue/recovery workers from the three cohorts described above. The overall aim of the project was to pool and de-duplicate data from three cohorts (FDNY, WTCHR, and GRC) for joint research on cancer incidence, latency and survival. The study design called for use of identical case ascertainment methods across the cohorts using state cancer registries; it also called for collaboration with the New York State Cancer Registry to manage and coordinate data pooling, and linking pooled data to 13 SCRs, ensure data security and confidentiality, and to harmonize data formats. There was also agreement to use a common exposure metric in the analysis.

This paper describes the processes involved with combining data across the three WTC-exposed cohorts and linking the pooled data with state cancer registries; and the strategies for overcoming administrative challenges. To our knowledge, studies that use pooled data do not typically provide a detailed description of their combining process. The transparency of this approach is important for a fuller understanding of the findings derived from analyses of WTC-exposure and health in our case, as well as, other endeavors that also use information combined from multiple sources.

## 2. Characteristics of the Study Cohorts

All three study cohorts—FDNY, WTCHR and the GRC—are programs funded by the NIOSH World Trade Center Health Program (WTCHP). The federal WTCHP supports 9/11-related research as well as medical monitoring and treatment for 9/11-certified medical conditions for rescue/recovery workers at clinical sites, including FDNY and ISMMS, and treatment for non-rescue/recovery workers at WTCHP clinical sites.

### 2.1. Fire Department of the City of New York (FDNY)

The FDNY cohort consists of all firefighters and emergency medical service providers who were employed by NYC and reported being at the WTC site at least one day between 11 September 2001 and 25 July 2002 (when the WTC site closed for FDNY recovery and clean-up efforts). WTC exposure information was obtained from self-administered health questionnaires beginning on 2 October 2002 and are completed during each routine health monitoring visit (12 to 18 months, even after retirement). Demographic and identifying information including sex, race, date of birth, full social security number and full name are obtained from FDNY employee records. The WTC-exposed FDNY cohort is 84.5% firefighters, 15.5% EMS and on 11 September 2001 had a median age of 41 for firefighters and 35 for EMS. Most of the firefighters were male (98%) compared with 80% for EMS [9]. Nearly all the firefighters (99%) were present at the WTC site sometime during the first two weeks after the disaster, with 16% present at the time of the towers’ collapse.

In 2011, FDNY published the first major cancer cohort study of WTC-exposed workers [4]. While FDNY now links to nine state cancer registries (Arizona, Connecticut, Florida, North Carolina, New Jersey, New York, Pennsylvania, South Carolina, and Virginia), at the time of this first study FDNY had received data only from Florida, Pennsylvania, North Carolina, New York, and Virginia. In that study, the WTC-exposed study population was defined as firefighters who were actively employed by FDNY on 1 January 1996, were employed for at least 18 months, and worked at the WTC site any day between 11 September 2001 and 25 July 2002 (further details are available [4]). The major study finding was that the overall Standardized Incidence Ratio (SIR) for all cancers was 1.10 (95% CI, 0.98–1.25) compared with the referent population. Several cancer-specific SIRs were significantly elevated, including melanoma, prostate, thyroid, and non-Hodgkin lymphoma. The reference age-specific rates were from the National Cancer Institute’s Surveillance Epidemiology and End Results (SEER) public use data released in 2013 [10]. For the current study, the FDNY final analytic cohort consisted of 16,221 individuals and included firefighters and EMS.

### 2.2. World Trade Center Health Registry (WTCHR)

The WTCHR was conceptualized in October 2001 to act as a registry for long-term research on individuals who were exposed to the 9/11 disaster. Recruitment and enrollment for the WTCHR were conducted by obtaining lists of potentially exposed persons via employers, unions, schools, and government agencies and by outreach and multi-media campaigns, which encouraged pre-registration through calling a toll-free number or pre- registering online [11]. The WTCHR was composed of four populations at risk including: rescue/recovery/clean-up workers and volunteers who had participated in these activities at the WTC site, Staten Island Recovery Center, or barges; residents of lower Manhattan (south of Canal Street) on 9/11; occupants of destroyed and damaged buildings or persons present south of Chambers Street on 9/11; and persons who were enrolled or working at schools in lower Manhattan on 9/11, including persons younger than 18 years on 9/11. Over 71,000 persons enrolled from 12 September 2003 to 15 November 2004, including 30,665 rescue/recovery/clean-up workers in this joint cohort, by completing an initial health survey in 2003–2004. The rescue/recovery/clean-up workers and volunteers were 80% male, with a median age of 42 year on 11 September 2001. The WTCHR has continued to monitor the health of this cohort via periodic health surveys, clinically based case-control studies, and matching with appropriate sources including cancer registries, hospitalization discharge records and death records.

The WTCHR has published two cancer surveillance reports in 2012 and 2016 [5,12]. Each report obtained linked cancer data from eleven state cancer registries (California, Connecticut, Florida, Massachusetts, New Jersey, New York, North Carolina, Ohio, Pennsylvania, Texas, and Washington) comprising 91% of all enrollees living in these states during the follow-up period. The 2012 study ended data collection at the close of 2008. When year of cancer diagnosis was limited to the last two years of follow-up, the SIR for all cancers combined was 1.14 for rescue/recovery/clean-up workers and 0.92 for non- rescue/recovery/clean-up workers, neither statistically significant. For rescue/recovery/clean-up workers, three cancer types were significantly elevated including prostate, thyroid, and multiple myeloma. In contrast, for non-rescue/recovery/clean-up workers, no specific cancers were elevated. In the 2012 study, there were no significant hazard ratios (HR) relative to the lowest exposure for cancers that had a significant SIR [12]. The 2016 WTCHR cancer report had three years of additional follow-up and limited cancer cases to those diagnosed from 2007–2011 [5]. In the 2016 report the overall SIRs for both rescue/recovery/clean-up workers (SIR = 1.11) and non- rescue/recovery/clean-up workers (SIR = 1.08) were statistically significant. Similar to the earlier report [12], for rescue/recovery/clean-up workers prostate and thyroid cancers were elevated, as was skin melanoma. Non-rescue/recovery/clean-up workers also had elevated prostate cancers and skin melanomas with additional elevation for non-Hodgkin lymphoma and female breast cancer. For the 2016 paper, a composite weighted score reflecting estimated total exposure to dust and debris during the nine-month rescue, recovery and clean-up period was developed and used [5]; however, there were no significant cancer associations with higher vs. lower levels of exposure. For the current study, the WTCHR final analytic cohort consisted of 29,372 WTC-site rescue/recovery/clean-up workers, some of whom are also members of other exposure groups in the WTCHR (e.g., residents and occupants of buildings in lower Manhattan).

### 2.3. General Responder Cohort (GRC)

The GRC consists of persons who were involved in WTC rescue/recovery and clean-up efforts and later enrolled in a predecessor of the WTCHP, which began in July 2002. Eligibility criteria for enrollment are based on having worked or volunteered in lower Manhattan, Staten Island, the Chief Medical Examiner’s Office, and barge-loading piers four hours or more between 11 September and 14 September 2001 or 24 h or more in September 2001, or 80 h or more from 11 September 2001 to 31 December 2001. Recruitment for this cohort consisted of outreach to unions and labor organizations and media campaigns. GRC members receive health monitoring visits every 12 to 18 months and treatment for WTC-certified conditions. The GRC includes participants from protective services (42%), construction (24%), buildings and grounds, maintenance and electrical, telecommunications and other installation and repair groups (10%), and other categories (19%) [6]. The median age on 9/11 was 38 years. High level exposure was 20%, categorized as 3% very highly exposed and 17% highly exposed.

Two GRC cancer incidence studies have been published. The first study was limited to 20,984 persons who enrolled in the WTCHP between July 2002 and 31 December 2008 and identified 552 individuals with cancer through linkages with four state tumor registries (Connecticut, New Jersey, New York, and Pennsylvania) [6]. In an analysis that restricted cancer cases to those that occurred at least six months after enrollment, and for any person enrolled during the observation period, a non-statistically significant SIR of 1.06 for all cancers combined was reported, but with significant SIRs for prostate and thyroid cancers. Multivariate models assessing the association between level of WTC exposure and cancer were suggestive of a trend but were not significant. The second study was an update of the earlier one, with an additional five years of follow-up through 2013 for residents of six states (Connecticut, Florida, New Jersey, New York, North Carolina, and Pennsylvania) [7]. With the additional follow-up, nearly twice as many cancers were diagnosed as in the earlier study (*N* = 1072); there was also an overall statistically significant SIR of 1.09 and significant SIRs for prostate and thyroid cancers. Unlike the prior study, the incidence of leukemia was also elevated. However, the associations between neither cancer overall nor prostate cancer and 9/11 exposures were statistically significant. For the current study, the GRC final analytic sample consisted of 33,427 individuals.

## 3. Creating a Combined WTC Rescue/Recovery Cohort

The process of creating a pooled dataset across these cohorts involved a number of steps, each described below.

### 3.1. Establish an Administrative Structure

A key component of the data pooling process was selection of an “honest broker”, one that would receive identifiable data, de-duplicate persons in more than one cohort, conduct matching with 13 state cancer registries and return a single, de-identified analytical file to researchers at WTCHR and FDNY for analysis. The New York State Cancer Registry (NYSCR), an essential partner in prior cancer surveillance work by all three cohorts, served as the “honest broker” for this project and their role was deemed a success story by CDC’s Program of Cancer Registries [13].

### 3.2. Identify State Cancer Registries for Linkage

The state cancer registries (SCRs) were selected based on the distribution of addresses on file of the rescue/recovery workers of the three cohorts and on our previous experiences performing cancer linkages [4,6,12]. Arizona, California, Connecticut, Florida, Massachusetts, New Jersey, New York, North Carolina, Ohio, Pennsylvania, Texas, Virginia, and Washington were selected.

Coverage in past cancer linkages ranged from 90 to 99% of cohort participants: 99% in FDNY cohort [14], 90% in the GRC [6], and 96% in the WTCHR cohort [5]. For all cohorts, the joint project would increase the number of state cancer registries rescue/recovery workers would be linked to and presumably result in increased coverage.

### 3.3. Obtain Required Institutional Review Board (IRB) Approvals and Legal Agreements

Before any data exchange could commence, the project required executing Data Use Agreements (DUAs) or in one case a Memorandum of Understanding (MOU) between parties involved in the project, applications to SCRs, and the completion of required Institutional Review Board (IRB) protocols and approvals (Figure 1). NYC DOHMH required that legal agreements be established with study partners in order to receive WTCHR data, including separate DUAs with the ISMMS and NYS DOH and a MOU with FDNY. It also required DUAs with all 13 SCRs (including the study partner NYSCR.)

The IRB at Albert Einstein College of Medicine served as the primary IRB, approving the study as minimal risk with a waiver of informed consent. It also served as the IRB for the FDNY via an IRB Authorization Agreement, and approved FDNY’s IRB study submission. IRB approvals were granted from the NYSCR and NYC DOHMH, and an IRB exemption was granted from the Icahn School of Medicine at Mt Sinai (ISMMS). Each of the 13 SCRs required a study submission to their IRBs, along with a supplemental SCR data request application for review and approval, all managed by NYC DOHMH. Applications were also submitted to the National Death Index (NDI), NYS and NYC Vital Records for the use of their mortality data.

Overall, it took two years and seven and a half months from the start of funding to complete all the legal agreements, applications and IRB approvals, and three years and 24 days until receipt of the final linked data.

The initial DUAs were based on a NYC DOHMH legal template with specific requirements for content. The process also consumed a substantial amount of administrative management time and effort and both to identify the appropriate parties for review and to obtain appropriate signatures. For example, the first seven and least cumbersome DUAs each required on average one phone call and 24 email communications and took approximately forty hours to successful completion. In some cases, legal representatives of participating agencies disagreed on language, requiring more time-consuming involvement of other officials in the negotiations.

Table 1 outlines the IRB and DUA associated with each state cancer registry in the study and the time for completion. Times shown indicate the date the initial DUA was emailed and the date the DUA was signed by both parties (“fully executed”). For DUAs between the NYC DOHMH and individual SCRs, the first DUA was submitted for approval to a SCR on 8 December 2016 and the last on 2 March 2017. The median time between submission of an initial DUA and execution of the DUA for the 13 SCRs was nineteen weeks, with a range from 6.5 to 114.5 weeks. Six states took longer than 20 weeks to approve the DUA: D (22.5 weeks), C (39.5 weeks), L (64.5 weeks), M (66.5 weeks), E (88 weeks), and I (114.5 weeks) (Table 1). We estimated that each IRB application required up to 40 h to prepare, not including responding to specific questions posed by IRB committees. The total estimated amount of time expended was approximately 680 h since the project required approvals from a total of seventeen separate IRBs (13 states, 3 cohort institutions and Einstein).

### 3.4. Join the Cohorts

Starting on 31 March 2017, each cohort delivered an encrypted file of their participants to the NYSCR including identifiable information on individual members. FDNY sent 16,221 records, WTCHR 29,372 and GRC 33,427 records, each consisted of name, birth date, address, phone numbers, social security number (to the extent available), race/ethnicity and fields allocated for other analytic variables such as exposure indicators. To create a joint file including all members of the combined Cohort for matching to cancer registries, the files first had to be standardized so that the analogous data items had the same variable names and formats. For example, social security numbers were standardized to nine characters, without dashes; names and addresses were formatted using upper case; and street numbers and names were concatenated, where appropriate. Challenges with the files were the disparity in completeness of names and addresses and the degree of missing information for some data elements, particularly social security number (Table 2), for which the GRC, over the years, first collected, then stopped collecting, then collected only last four digits.

The first linkage for de-duplication identified 302 individuals in both FDNY and GRC cohort records (primarily retired FDNY). The NYSCR reviewed the matched records manually and consolidated each matched pair into a single record. The two files (GRC and FDNY) with unmatched records were combined, and the consolidated single records were added resulting in a joined dataset of 49,346 unique records, each with an indicator variable of the cohort source(s) and the original cohort ID associated with each record. Then the combined GRC/FDNY file was matched to the WTCHR file, using LinkPlus. In this de-duplication there were 9616 matched records (due to persons in the WTCHR and in either GRC and/or FDNY). As before, the NYSCR reviewed the matched records manually and consolidated each matched pair into a single record. The two files with unmatched records (GRC/FDNY and WTCHR) were combined, and the consolidated single records were added, resulting in a joined final dataset of 69,102 unique records, each record with an indicator variable of the cohort source(s), a new unique WTC joined ID. This final dataset created the combined Cohort.

During the joining of the cohorts, NYSCR applied decision rules developed by the joint project team. For matched pairs of records, the rules included retention of the more complete information for each matching data variable such that complete versus partial values, and present versus missing values were selected for retention. For example, complete social security numbers, names, and date of birth values were preferred and retained over missing or partial values. When more than one address and/or phone number was provided, the consolidated record included all values. WTC exposure information was included in its entirety from the matched record that had the earliest enrollment date. It should be noted that records from FDNY typically had the earliest enrollment date so that if an FDNY record was included, exposure information was based on FDNY information. The diagram below shows the overlapping records within the joined files (Figure 2).

The number of records that overlapped between WTCHR and GRC only was 6377 (22.0% of WTCHR), between WTCHR and FDNY only was 3154 (11% of WTCHR), and between GRC and FDNY only was 217 (0.9% of GRC) and between WTCHR, GRC, and FDNY was 85 (0.3% of 69,102 total records). The joining of the cohorts resulted in an increased sample for analysis beyond what was otherwise available for the FDNY by 77%, for WTCHR by 57%, and for GRC by 52% for analysis.

As part of creating the joined file, an agreed-upon flag variable was created to indicate membership of an individual in the three cohorts in the combined Cohort file. The exclusive three categories included (a) member of FDNY regardless of membership in GRC or WTCHR (*n* = 16,221); (b) a member of GRC, but not a member of FDNY, but could be member of WTCHR (*n* = 33,125); (c) member of the WTCHR but not the GRC or the FDNY (*n* = 19,756). This categorization resulted in masking the identity of individuals who were members of the GRC and the WTCHR.

### 3.5. Match with Each State Cancer Registry

NYSCR matched the combined Cohort file to the NYSCR consolidated tumor file and included the project-specific WTC_ID for each record. NYSCR also linked the combined Cohort file with the 12 other participating state cancer registries (Table 1). After the combined Cohort file was created, responsible personnel at the targeted state cancer registries were contacted concerning secure methods for data exchange and for information on state-specific format requirements for the combined Cohort file. As IRB approvals and executed DUAs were obtained for each state, NYSCR reached out to each state to request a data linkage of the joined file to the SCR. A file was submitted in November 2017 to the first SCR (Table 1). Separate encrypted files containing combined Cohort were prepared for each state and transferred to each participating cancer registry. The files containing matched cancer cases returned by the states varied by type of data set including SAS, excel, .txt, and .pdf. The matching process was completed when the final file of matched results was returned from the last SCR in September 2019. Thirteen specific data sets were provided, one from each SCR, with the match results (Table 3). All states included matches of cancer cases diagnosed between 2000 and 2015. Although each state received the entire combined Cohort file regardless of the state of residence of the combined Cohort member that is the number of persons potentially residing in each state, the number of tumor records varied from a low of 193 to a high of 51,196; 20 matches were still identified in the state “E” with the fewest residents.

The combined files from each state yielded 7894 tumors, which had year of diagnosis as early as 1975. Overall, 434 tumors were reported by more than one SCR. The duplicate records were then manually consolidated by NYSCR. Most duplicates were from New York and New Jersey (201 for NY and 159 for NJ). When one of the duplicate records was from NYS then the NYSCR record was retained; otherwise, the record with the earlier date of diagnosis was retained. The incidence data set consisted of single records for each tumor identified by a SCR among persons matched to the combined Cohort file and contained tumor information. NYSCR created a final joined data set with tumor information reported between 2002–2015 (*n* = 6046) and transferred these data to FDNY and WTCHR (Figure 3). The combined Cohort increased the sample of tumors beyond what any single cohort had for analysis such as 58% more tumors than what GRC alone had for analysis. However, the final number of tumor cases could be fewer for any given inclusion or exclusion criteria.

## 4. WTC Exposure for Combined Cohort Data

The five previously published cancer studies from these cohorts used different definitions of WTC exposures, based on the best information available for each cohort at the time [8]. There was wide variation in the activities designated as “highest exposure” for rescue/recovery/clean-up workers, ranging from the FDNY definition, which included all individuals who arrived at the WTC site on the morning of 9/11 before the buildings collapsed, WTCHR and GRC designations which incorporated specific information about working on the pile, length of time working at site (e.g., >90 days for both WTCHR and GRC), and being caught in the dust/debris cloud at time of the towers’ collapse.

Three exposure constructs are helpful in organizing available exposure information for rescue/recovery/clean-up workers: (a) delineating exposure levels on 9/11; (b) specifying time periods worked between 11 September and 25 June 2002; and (c) recording tasks performed. A key assumption was that arriving on site during the morning of 9/11 would place early rescue/recovery workers on-site at the time of towers’ collapse, with ensuing entrapment in the dust and debris cloud. Four hierarchical circumstances of exposure were created:Type A: Presence in lower Manhattan on 9/11 and heavy exposure to dust from the cloud.Type B: Presence in lower Manhattan, work on the pile on 9/11 resulting from the collapse of WTC towers on 9/11, not present during period of heaviest exposure to dust from the cloud.Type C: Presence in lower Manhattan on 9/11, no pile work on 9/11, not present during period of heaviest exposure to dust from the cloud.Type D: Not present in lower Manhattan on 9/11.

For exposure Type A, FDNY distinguished between being present or not at the time of towers’ collapse, WTCHR and GRC asked directly about being in the dust/debris cloud. Those who were not present during the towers’ but who worked on the pile on the WTC site on 9/11 were included in Type B. In this instance, WTCHR asked directly about working on the pile on 9/11. However, GRC included those who did not have the full dust cloud exposure on 9/11 and FDNY asked whether the firefighters engaged in rescue activities such as fire suppression or rescue/recovery at the site on 9/11. Type C would require being present in the vicinity of the WTC site on 9/11, but none of the other criteria of Type A or B. Type D would not be present at the WTC site on 9/11. The three cohorts varied considerably on the distribution of this category with only 35% of FDNY arriving at the WTC site after 9/11, 60% for WTCHR and 50% of GRC, likely representative of the occupational differences between members of the cohort and their responsibilities on and after 9/11.

A second exposure construct estimates the burden of work on the effort via earliest arrival time, time periods worked (e.g., 11 September 2001, 12 September, 13 September–17 September, 18 September to 30 June 2002) and total number of days worked, but not all entities had the same information available, especially number of days worked on pile or site and which days or period of time they worked. As a result of pooling data from the three cohorts, time of arrival was more evenly distributed than for individual cohorts. For instance, 62% of the FDNY cohort reported a 9/11 start date (15% of the entire pooled data) and 4.7% who arrived after 17 September. After pooling of FDNY with the GRC and WTCHR, 39% arrived on 11 September and 21% after 17 September, with 20% for each of the other two time periods (12 September and 13 September–17 September).

FDNY and WTCHR obtained information about tasks performed while working on the pile. These included firefighting (WTCHR), fire suppression at site (FDNY), search and rescue (WTCHR), rescue/recovery at site (FDNY), hand digging or bucket brigade (WTCHR), digging at site (FDNY), welding, steel-cutting/torch operation (WTCHR), welding or steel cutting at site (FDNY), heavy equipment operation (WTCHR), and light construction (WTCHR). GRC members selected an activity code (from a list: body bag work, firefighter, bucket brigade, industrial hygienist, cable/installation/repair/splicing (excluding work performed in manholes), morgue work, cable installation/repair/splicing (including work performed in manholes), police officer, canteen services, perimeter security, counselor, sanitation workers, custodian, search and rescue, dog handlers, sifting (excluding conveyer belt), dust suppression, sifting (including conveyer belt), EMT, towing, escorting, truck loading/unloading, excavation/confined space work, truck routing) that best described what they were doing in September, October, November–December 2001 and January–June 2002. WTCHR also obtained information on work activities by time periods (11 September, 12 September, 13 September–17 September, 18 September–31 December after 31 December 2001), while FDNY obtained this information only for 11 September, 12 September, 13 September, 14 September 2001 to 24 September 2001 and 25 September 2001 to 25 July 2002 and months from September to July.

The summary exposure data for the pooled cohort therefore includes (a) dust/debris exposure on of 9/11 (Type A–D); (b) date of arrival (11 September 2001, 12 September 2001, 13 September–17 September 2001, after 17 September 2001); and (c) ever performed tasks on pile (yes/no). This information is included for every individual in the pooled data file.

## 5. Demographic Characteristics and WTC-Exposure of Combined Cohort

The pooled cohort resulted in a more balanced demographic representation than single cohorts (Table 4). For instance, FDNY was over 99% male and 94% non-Hispanic white on 9/11, in contrast to the combined Cohort with 15% female and 68% non-Hispanic white. In addition, the distribution of Date of Arrival, a key surrogate for intensity of WTC exposure was more balanced, especially with 18% assigned to a potential reference group of arriving on September 18 or afterward.

## 6. Discussion

There is considerable scientific merit in joining three distinct cohorts of rescue/recovery workers for the common goal of assessing the association between cancer incidence and exposure to pollutants during and following the WTC disaster. This manuscript provides a detailed description of the process, and the great efforts, required for the realization of that merit. First, substantial administrative preparation and persistence were required. Second, complexity was added by fundamental differences across the study designs, e.g., closed versus open enrollment, and WTC exposure level (on 9/11, after 9/11) were difficult to harmonize. Regardless of these initial differences, however, the final outcome of the joining process was a greatly strengthened resource for monitoring the impact of 9/11 exposures on cancer risk among rescue and recovery workers and volunteers. It can be noted also that the total number of pooled and de-duplicated combined Cohort of 69 thousand approaches the estimated population at risk of WTC rescue/recovery workers of 91 thousand [1].

Although the central task for combining the cohorts was merging the members, eliminating duplicates to create the combined Cohort and matching the Cohort with cancer registries, the overwhelming consumption of resources and effort went into building of an administrative infrastructure before that process could even begin. An earlier report on matching a large cohort of jet engine manufacturing workers (225,000) with over 30 SCRs estimated it required around 400 h to complete state IRB and applications [15]. As we also experienced, the prior study [15] reported a wide variation in the responsiveness and efficiency of SCRs on application requirements and follow-up of matching and provision of cases, which some states never completed before the researchers abandoned the effort. For this project, there was also a wide range in time from application completion to receipt of matched data. In addition to the SCRs’ matching, our project also required the completion of multiple data use agreements across the administrative homes of each cohort (in addition, of course, to IRB approval within each participating entity that received the data). Typically, a grant-funded project with a three- to five-year funding period would expect that the administrative tasks would be completed in the first year. In the case of the combined Cohort, it required more than three years of the project’s funding period before data were available for analysis, although this included the actual data linkages. Even so it was fortunate that the NYSCR served as the honest broker, bringing their substantial expertise for both merging of the cohorts and the interacting with the twelve other SCR’s that had been identified in previous matches as comprehensive sources of cases for about 95% coverage. Future pooling of cohorts may avail of potential efficiencies in matching with SCRs, such as Virtual Pooled Registry Cancer Linkage System (VPR-CLS), by which linkages are completed centrally under the guise of the North American Association of Cancer Registries (NAACCR.org). Also, the vagaries of this project were acknowledged by the project officers of the sponsor institution, the National Institute for Occupational Safety and Health [16].

The sample size (*N* = 69,102) of the combined Cohort is substantially larger than what individual cohorts have available for analysis. This increase amounts to 77% for FDNY, 57% for WTCHR and 52% for GRC. As a result of pooling, there are numbers of distinct advantages for assessing 9/11- related cancer, for instance, to satisfy sample size requirements for change point analysis to estimate latency or to conduct survival analysis that includes cancer stage. In addition, there is increased capability for assessing the association of 9/11 exposure with lower incidence cancers that have been identified as of concern in prior reports such as multiple myeloma [17] or kidney cancer [5] due to improved statistical power. The pooled cohort also has the potential of increasing the scope of 9/11 exposures that could be used for internal comparisons. For instance, a very large proportion of FDNY responders arrived on 9/11 in comparison to the other cohorts where arrival and duration of exposure had a greater spread. After pooling the data, we have the potential of achieving greater efficiency with more balanced numbers between exposure groups, especially with a larger reference such as time of arrival after 17 September 2001, producing more precise estimates [18]. Other than sample size increase, the pooling of the cohorts also takes advantage of the heterogeneity of the different cohorts, thereby increasing the representation of different populations of risk with a unified file of common elements.

The project goals set out by Boffetta, et al. [8], have largely been achieved, including agreement on a common set of state cancer registries with standardized elements for matching. Also, the combined Cohort study group can select a reference population for the study population using a clearly stated rationale and for a specific set of years. Reference populations could be modified according to analytic objectives, but with increased power to detect signals for less common cancers. Given that the prior reports on cancer as discussed above used very different 9/11 exposure schemes, the combined Cohort cancer analysis will use a common framework for exposure definitions. These strategies for enhanced analytic capabilities offset the limitations created by combining cohorts, which includes such things as differences in enrollment strategies, variation in level of missing information for matching, and loss of detail of WTC-exposure information.

## 7. Conclusions

The consolidation of data from three WTC-exposed rescue/recovery worker cohorts that have previously reported results on cancer association with WTC exposures has resulted in a merged file of over 69,000 unique individuals and formed the combined Cohort. This resource for future research has sufficient sample size for a number of hypothesis testing in observational studies of WTC exposures in relation to outcomes, such as rare cancers and mortality. In addition, the combined Cohort has an infrastructure in place, including a collaborative scientific team, an honest broker (NYSCR), DUAs and IRB protocols that require minimal update effort at regular intervals of three to five years for monitoring of cancer incidence in WTC-exposed rescue/recovery workers. This paper, which details the methods for combining or pooling data from separate cohorts, assists the interpretation of findings from pooled data, allows for reproducibility, identifies limitations, and informs comparability of findings between studies.

## Figures and Tables

**Figure 1 ijerph-18-01386-f001:**
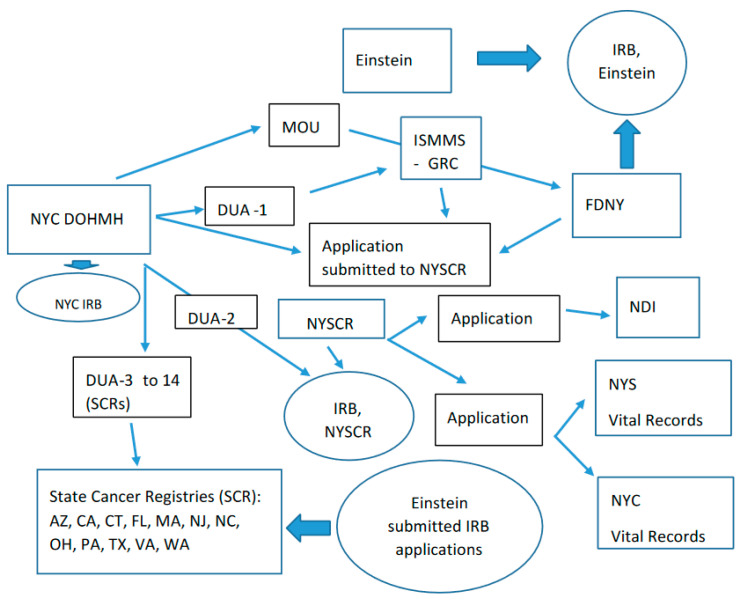
Administrative agreements across study entities. Abbreviations: IRB, Institutional Review Board, MOU, Memorandum of Agreement, ISMMS, Icahn School of Medicine at Mt Sinai, GRC, General Responder Cohort, FDNY, Fire Department of New York, NYC DOHMH, New York City Department of Health and Mental Hygiene, DUA, Data Use Agreement, NYSCR, New York State Cancer Registry, NDI, National Death Index, NYS, New York State, NYC, New York City, SCR, State Cancer Registry, AZ, Arizona, CA, California, CT, Connecticut, FL, Florida, MA, Massachusetts, NJ, New Jersey, NC, North Carolina, OH, Ohio, PA, Pennsylvania, TX, Texas, VA, Virginia, WA, Washington.

**Figure 2 ijerph-18-01386-f002:**
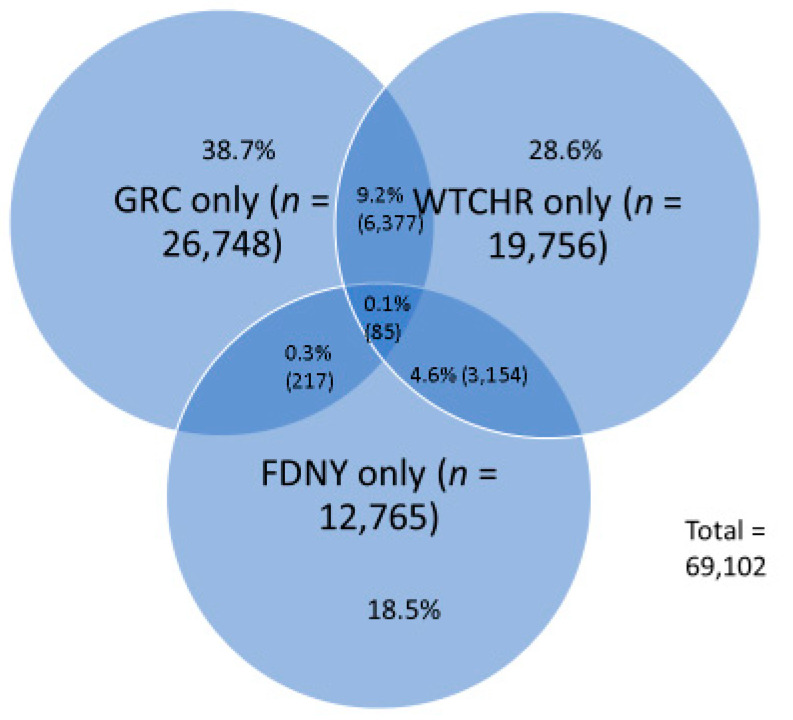
Overlap between the three cohorts after de-duplication and combining.

**Figure 3 ijerph-18-01386-f003:**
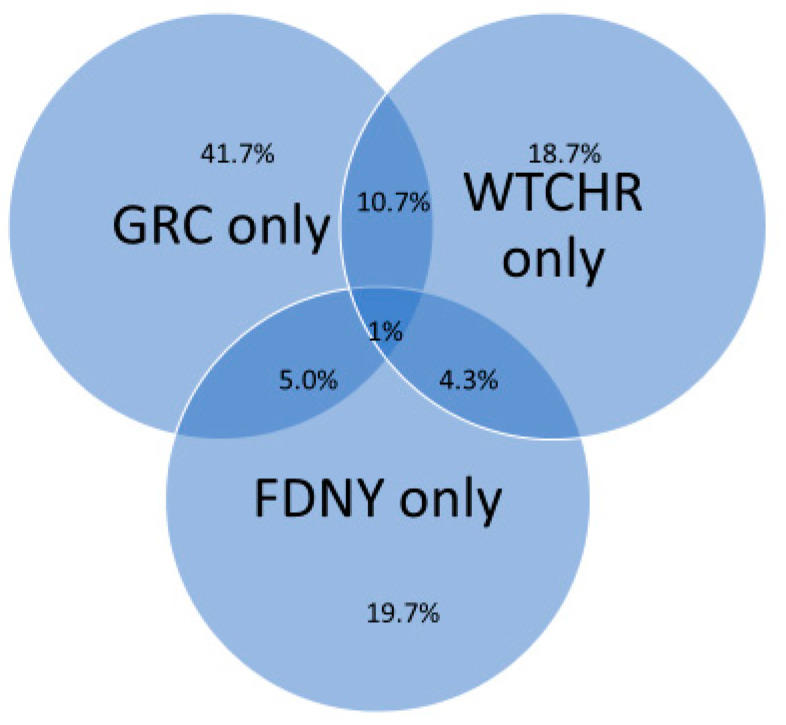
Distribution of cancer cases across the three cohorts after joining and matching with state cancer registries. These are cancer cases reported between 2002–2015 for the 64,174 rescue/recovery workers who reported at last known address in one of the 13 states with which data was linked. *N* = 6046.

**Table 1 ijerph-18-01386-t001:** Institutional Review Board (IRB) and Data Use Agreements (DUA) submissions and dates of approval and linkage dates of submission and date of return by state cancer registry.

State	Initial IRB		DUA		Linkage		Date Returned
	Date Submission	Date Approval	Date Sent to State Cancer Registries (SCR)	Date Signed/Executed	Data Available Started	Date Submission	
A	28 August 2017	29 January 2018	14 December 2016	13 March2017	1988	2 July 2018	24 August 2018
B	11 September 2017	6 March 2018	30 January 2017	16 March 2017	1982	20 April 2018	26 April 2018
C	13 April 2018	3 July 2018	30 January 2017	3 November 2017	1985	3 July 2018	9 August 2018
D	5 September 2017	25 September 2017	8 February 2017	17 July 2017	1990	8 November 2017	14 December 2017
E	11 May 2017	24 October 2017	30 January 2017	9 October 2018	1992	16 January 2019	2 May 2019
F	10 August 2017	29 November 2017	30 January 2017	16 March 2017	1990	2 January 2018	26 February 2018
G	8 September 2017	24 October 2017	30 January 2017	12 May 2017	1991	8 November 2017	3 January 2018
H	3 August 2017	18 January 2019	30 January 2017	14 April 2017	1979	2 July 2019	24 September 2019
I	20 February 2018	7 February 2019	30 January 2017	12 April 2019	1981	24 April 2019	25 June 2019
J	8 November 2017	9 April 2018	8 December 2016	24 April 2017	1973	9 May 2018	17 May 2018
K	1 September 2016	28 October 2016	2 March 2017	19 April 2017	1976	18 January 2018	12 February 2018
L	23 October 2017	2 February 2018	30 January 2017	27 April 2018	1995	30 April 2018	31 May 2018
M	8 September 2017	21 September 2017	28 February 2017	8 June 2018	1992	26 June 2018	30 August 2018

**Table 2 ijerph-18-01386-t002:** Summary of partial and missing information for data files provided to NYSCR for pooling.

Data Element	FDNY	GRC	WTCHR
Name *	0%	0%	0.3%
Birth date	0%	0%	0.4% missing 0.01% partial
Address	0%	1.7%	0.1%
SSN	0%	52.3% missing 19.5% partial	24.6% missing 8.0% partial
Race **	7.1%	31.3%	16.3%
Hispanic ethnicity	0.4%	20.6%	1.8%

* Either missing or only one letter. ** Either missing or reported as ‘other’. Abbreviations: NYSCR, New York State Cancer Registry, FDNY, Fire Department of New York, GRC, General Responder Cohort, WTCHR, World Trade Center Health Registry

**Table 3 ijerph-18-01386-t003:** Tumor matches for 13 state cancer registries.

State	No. of Mmatches Regardless of Year of Diagnosis	No. in WTC combined Cohort Who Resided in States Based on the Last Known Residence
A	41	791
B	19	330
C	131	1284
D	36	415
E	20	193
F	108	777
G	23	236
H	928	5588
I	236	2668
J	58	575
K	6239	51,196
L	25	373
M	30	248
Total	7894 **	64,174 *

* Matches with 13 state central cancer registries covered 93% (64,174/69,102) of the cohort members, based on the members last known state of residence. ** When limited to reporting date, 2002 to 2015, there are 6046 tumors.

**Table 4 ijerph-18-01386-t004:** Characteristics and WTC exposure of the combined cohort (*N* = 69,102).

Variable	No.	(%)
Socio-demographics		
Age on 9/11, year		
<18	165	(0.2)
18–29	10,375	(15.0)
30–39	26,417	(38.2)
40–49	21,510	(31.1)
50–59	8438	(12.2)
≥60	2176	(3.1)
Missing	21	(0.03)
Hispanic/Latino		
Yes	10,724	(15.5)
No	52,007	(75.3)
Missing	6371	(9.2)
Race		
White	47,166	(68.3)
Black	6506	(9.4)
American Indian or Alaska Native	199	(0.3)
Asian or Pacific Islander	1274	(1.8)
Other	3405	(4.9)
Unknown	10,552	(15.3)
Sex		
Male	58,371	(84.5)
Female	10,731	(15.5)
Smoking status at enrollment		
Current	10,419	(15.1)
Former	16,157	(23.4)
Never	40,634	(58.8)
Missing	1892	(2.7)
Year of enrollment		
2001–2004	49,257	(71.3)
2005–2016	19,845	(28.7)
Exposures at WTC site		
Date of arrival		
11 September 2001	24,737	(35.8)
12 September 2001	12,717	(18.4)
13–17 September 2001	13,034	(18.9)
≥18 September 2001	12,700	(18.4)
Not at the WTC site	5095	(7.4)
Missing	819	(1.2)
Dust/debris exposure on 9/11		
Type A	13,418	(19.4)
B	12,287	(17.8)
C	4901	(7.1)
D	34,160	(49.4)
Missing	4336	(6.3)
Ever performed tasks on pile		
Yes	25,250	(36.5)
No	42,715	(61.8)
Missing	1137	(1.6)

## Data Availability

Data from study may be obtained from the Contact Principal Investigator (CBH) upon reasonable request after approval by the Steering Committee for “Incidence, Latency, and Survival of Cancer Following World Trade Center Exposure” (NIOSH Cooperative Agreement U01 OH011932) in accordance with the study’s official Data Sharing Plan.

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
