# Peer review of "Combining Three Cohorts of World Trade Center Rescue/Recovery Workers for Assessing Cancer Incidence and Mortality"

_ijerph, 2021, doi:10.3390/ijerph18041386_

Round 1

Reviewer 1 Report

This is a well written article that is unusual in describing, in some detail, the process of creating a single usable data base from three different sources.  Frankly, the only problem I see with the article is that the authors do not describe the nature and purpose of the article at the outset.  They really need to do that so that readers will know what they are getting into.  In particular, the authors need to state, at the beginning, WHY they are describing the complex processes involved in combining three data sets.  They might also, at the end, reiterate how they believe readers might use the article in their own work.  

There are some typos and misused words but I believe that would be better handled during editing.  

Author Response

1/29/2021

To Editor of IJERPH:

Response to reviewers on manuscript “Combining Three Cohorts o f WTC Rescue/Recovery Workers for Assessing Cancer Incidence and Mortality”

Reviewer 1:

Comments and Suggestions for Authors

This is a well written article that is unusual in describing, in some detail, the process of creating a single usable data base from three different sources.  Frankly, the only problem I see with the article is that the authors do not describe the nature and purpose of the article at the outset.  They really need to do that so that readers will know what they are getting into.  In particular, the authors need to state, at the beginning, WHY they are describing the complex processes involved in combining three data sets.  They might also, at the end, reiterate how they believe readers might use the article in their own work.  

There are some typos and misused words but I believe that would be better handled during editing.  

Response: We thank reviewer 1 for suggestion to give the reader a better idea of why read this paper. We have added several sentences to last paragraph of the introduction providing more explicit reason for having description of methods used for pooling data (lines 119- 124) and a sentence in the conclusion that also says how the methods paper has utility in our own work (lines 559 – 562)

We have made additional edits to fix some typos and misused words.

Reviewer 2 Report

In this manuscript, the authors describe the process and the effort of merging the datasets from the three WTCHP cohorts for assessing cancer incidence and mortality. They detailedly discussed how they obtain IRB approvals, match cancer cases with SCRs, and define WTC exposures. The merged data will provide greater statistical power to investigate the association between WTC exposures and cancer incidence. While the manuscript is well-written, I have some suggestions.

Introduction, line 69-71, authors mentioned “the results have been consistent, with effect estimates for all cancers from the most recent studies ranging from 6 to 14 % in excess of background rates”. Any reference to support this sentence?

Line 115, full form of the abbreviation SCR is missing in the introduction, although it was explained in the abstract. It is better to provide a list of abbreviations at the end of the paper since there are many abbreviations in the text.

Line 123, it is a little bit confuse that why WTCHR is in the parenthesis, abbreviation?

Line 147, (0.98-1.25) is the 95% confidence interval (CI) of the SIR 1.10, right? However, the CI contains 1 which means the SIR is not significant at 0.05 significance level. Why do you think this is important to mention this non-significant result here? And it is better to add “95% CI“ into the parenthesis.

The percentages displayed in line 350-352 are different from the percentages showing in Figure 2. It is confusing that you present the overlapping percentages of the joint data in the Venn diagram but interprets them in a sentence using a different way.

Section 3.5, does the combined file contain any information about patients with multiple primary cancers and also cancer-related biomarkers?

The paper seems only to describe the process of merging datasets. Do you have any preliminary data about the cancer incidence and mortality for the merged data?

As we know, subjects in FDNY cohort are mainly males. So, what’re the demographic characteristics of the subjects look like for the merged data (e.g., age on 9/11, age of cancer diagnosis, sex, smoking status,etc)?

Author Response

To Editor of IJERPH:

Response to reviewers on manuscript “Combining Three Cohorts o f WTC Rescue/Recovery Workers for Assessing Cancer Incidence and Mortality”

Reviewer 2:

Comments and Suggestions for Authors

In this manuscript, the authors describe the process and the effort of merging the datasets from the three WTCHP cohorts for assessing cancer incidence and mortality. They detailedly discussed how they obtain IRB approvals, match cancer cases with SCRs, and define WTC exposures. The merged data will provide greater statistical power to investigate the association between WTC exposures and cancer incidence. While the manuscript is well-written, I have some suggestions.

Introduction, line 69-71, authors mentioned “the results have been consistent, with effect estimates for all cancers from the most recent studies ranging from 6 to 14 % in excess of background rates”. Any reference to support this sentence?

Response: We thank reviewer 2 for suggestions on how to improve the paper. The reference for the statement on effect estimates was added.

Line 115, full form of the abbreviation SCR is missing in the introduction, although it was explained in the abstract. It is better to provide a list of abbreviations at the end of the paper since there are many abbreviations in the text.

Response: We have added the full form of SCR in the text. We will leave it up to the journal editors as whether to have all definitions of abbreviations in a separate place. It seems this would mean removal of the definitions in the body of the paper which could also be confusing.

Line 123, it is a little bit confuse that why WTCHR is in the parenthesis, abbreviation?

Response: WTCHR was removed which was an error.

Line 147, (0.98-1.25) is the 95% confidence interval (CI) of the SIR 1.10, right? However, the CI contains 1 which means the SIR is not significant at 0.05 significance level. Why do you think this is important to mention this non-significant result here? And it is better to add “95% CI“ into the parenthesis.

Response: We think that this is important because the confidence interval is narrow and the estimate is similar to what was found based on data from other 9/11 rescue/recovery cohorts. We have added 95% CI.

The percentages displayed in line 350-352 are different from the percentages showing in Figure 2. It is confusing that you present the overlapping percentages of the joint data in the Venn diagram but interprets them in a sentence using a different way.

Response: We thank reviewer for seeing this confusion and have revised the Venn diagram to show both the n’s in each section of the diagram and the percentages. The text in lines 356-359 have been revised to be more in accordance with what is depicted in the Venn diagram.

Section 3.5, does the combined file contain any information about patients with multiple primary cancers and also cancer-related biomarkers?

Response: The matched records included all the tumors for any individual matched.

The paper seems only to describe the process of merging datasets. Do you have any preliminary data about the cancer incidence and mortality for the merged data?

Response: Information on cancer incidence and mortality will be reported in separate manuscripts in which the statistical methods for deriving this information can be described.

As we know, subjects in FDNY cohort are mainly males. So, what’re the demographic characteristics of the subjects look like for the merged data (e.g., age on 9/11, age of cancer diagnosis, sex, smoking status,etc)?

Response: We thank the reviewer for the excellent suggestion. We have added a table with demographic exposure information on the pooled data. This will be useful for the individual papers as a reference for the sample used in these papers after appropriate exclusions are applied to the entire sample of 69,102.